# Axial Impact Load of a Concrete-Filled Steel Tubular Member with Axial Compression Considering the Creep Effect

**DOI:** 10.3390/ma12193134

**Published:** 2019-09-26

**Authors:** Tao Lan, Guangchong Qin, Jinzhao Zhuang, Youdi Wang, Qian Zheng, Min Ding

**Affiliations:** 1CSIC International Engineering Co. Ltd. & CSIC Co. Ltd., Beijing 100121, China; qd_lantao@163.com (T.L.); qinguangchong@126.com (G.Q.); 2College of Water Resources and Civil Engineering, China Agricultural University, Beijing 100083, China; zhuangjz@cau.edu.cn (J.Z.); zhengq1996@cau.edu.cn (Q.Z.); 3Beijing Fangshan Future City Investment CO., Beijing 102400, China; wangyoudi@cau.edu.cn

**Keywords:** concrete-filled steel tubular member, axial compression, numerical simulation, creep, dynamic response, factor

## Abstract

The dynamic loads acting on concrete-filled steel tubular members under axial impacts by rigid bodies were studied herein by FEM. The whole impact process was simulated and the time history of the impact load was obtained. The effects of eight factors on the axial impact load were studied; these factors were the impact speed, mass ratio, axial pressure ratio, steel ratio, slenderness ratio, concrete strength, impact position, and boundary conditions. Besides this, the effects of concrete creep on the impact load were also considered by changing the material parameters of the concrete. The results show that axial impact load changes with time as a triangle. The peak value of impact load increases and the impact resistance improves with the growth of the axial pressure ratio, steel ratio, slenderness ratio, and concrete strength after creep occurs. As the eccentricity of the axial impact acting on a concrete-filled steel tubular member increases, the peak value of the impact load decreases. The enhancement of constraints at both ends of the member can improve the impact resistance. The creep reduction coefficients for the peak axial impact load of a concrete-filled steel tubular member under axial compression and considering the creep effect over 6 months and 30 years are 0.60 and 0.55, respectively. A calculation formula for the peak value of impact load was suggested based on the existing formula, and its accuracy was proved by finite element calculation in this study.

## 1. Introduction

A concrete-filled steel tubular member is a combined member which is produced by filling a thin-walled steel tube with concrete. It is more and more widely used in actual projects as the major load-bearing member due to its high strength and good plasticity. Under the action of axial compression at the service stage, the core concrete will suffer creep as time goes on; this will cause the cooperative work of the core concrete and steel tube to be broken and cause the redistribution of their internal forces. Generally, the core concrete stress will become relaxed and the steel tube stress will increase. If the member encounters impact load at this time, the steel tube will probably enter into a plastic state and have local buckling. 

Concrete creep was firstly observed by Furlong [1] in 1967. Koh and others [2] adopted the BP model to calculate the deformation of a reinforced concrete beam under static load and repetitive load. Sapountzakis [3,4,5] analyzed the dynamic responses of a reinforced concrete beam and slab structure with creep effects by considering the internal force and deformation of slabs and the axial force and deformation of beams. The results showed that the maximum free vibration displacement of the reinforced concrete beam and slab structure after concrete creep occurred was 2.5 times that when there was no creep, and the static displacement was 4.51 times that when there was no creep. Zhou Yao [6] and Mao KunZhong [7] discovered that concrete creep has a big influence on the natural vibration frequency and dynamic responses of concrete-filled steel tubular members, especially during the early stage of creep.

The research on the dynamic performance of concrete-filled steel tubular members started relatively late. So far, studies on the seismic performance of concrete-filled steel tubular members are comparatively exhaustive. The existing studies on the dynamic responses of concrete-filled steel tubular members under impact load, such as the references [8,9,10,11,12,13,14], are mostly based on axial impact experiments to analyze the dynamic responses, and their load cases are relatively simple. There is little in the literature about the axial impact load of concrete-filled steel tubular members under axial compression, especially about that after concrete creep occurs. 

This paper explains a finite element model established in ABAQUS software to simulate the whole process when a concrete-filled steel tubular member with creep effect bears axial impact. The axial impact load is discussed by setting different values for the impact speed, mass ratio, axial pressure ratio, steel ratio, slenderness ratio, concrete strength, impact position, and boundary conditions. The peak reduction coefficient of the axial impact load due to concrete creep is concluded, and a calculation formula for the axial peak impact load of a concrete-filled steel tubular member under axial compression and considering the creep effect is suggested based on the existing formula. This will provide a theoretical basis for the impact resistance design of concrete-filled steel tubular members.

## 2. Basic Parameters and the Numerical Model 

### 2.1. Geometric Model and Impact Parameters

The geometric model of a concrete-filled steel tubular member under impact load is as shown in Figure 1. The detailed parameters of the basic specimen are as shown in Table 1. 

As shown in Table 1, *D* and *t* are the external diameter and wall thickness of the steel tube, respectively, and *L* is the member length. *α_s_* is the steel ratio, which can be calculated with the formula $\alpha_{s}=\frac{A_{s}}{A_{c}}$, and the slenderness ratio is λ=4L/D. According to code GB50936-2014, the design value of the axial compression capacity of a concrete-filled steel tubular member is *N_u_* = 1398.2 kN for those geometric dimensions given in Table 1. Because the concrete-filled steel tubular member is generally in the elastic condition at the service stage when it carries an axial compression load, the maximum axial pressure ratio *n = N/N_u_* set in this study was no more than 0.6, and the value of axial force *N* for an axial pressure ratio of 0.3 is given in Table 1. The mass ratio *x = m/M* was also studied, where *m* and *M* are the impact block mass and the member mass, respectively. The member mass *M* is constant and *M* = 180 kg in this paper. Table 1 shows the value of impact block mass for a mass ratio of *x* = 0.5. *v* is the velocity of the impact block.

The effects of the eight factors of impact speed, mass ratio, axial pressure ratio, steel ratio, slenderness ratio, concrete strength, impact position, and boundary conditions on the axial impact load were analyzed. The values of the parameters given in Table 1 are the basic parameters; that is to say, when studying a factor’s effect, a specific parameter was changed, but other parameters were set as shown in Table 1. The parameters and values that were changed are given in later sections.

### 2.2. Material Properties

Q235 steel was used for the steel tube. A linear hardening constitutive model was used to simulate the steel tube; the elasticity modulus of the steel tube was *E_s_* = 210 GPa, the hardening modulus was *E_t_ = *0.01*E_s_* = 2.1 GPa, the ultimate tensile strength was *f_u_* = 400 MPa, and the ultimate strain was *ε_u_* = 0.089. 

C40 concrete was used for the core concrete. HAN Linhai’s constitutive model for core concrete in a concrete-filled steel tube [11], which considers the confinement effect on the concrete’s strength, was used to calculate the ultimate strength to estimate whether or not the member was destroyed during the impact process. The creep effect in the core concrete at different ages is equivalent to change in its elasticity modulus; the equivalent elasticity moduli for the core concrete of a concrete-filled steel tubular member considering the creep effect were calculated according to the references [15,16] and were as shown in Table 2.

The concrete damaged plasticity model (CDP) in ABAQUS was used to simulate the material nonlinearity of the core concrete. A damage factor was introduced in the CDP, and stiffness degeneration and unrecoverable deformation were considered when unloaded. According to the stress vs. strain curve of concrete under uniaxial compression, the stress often linearly increases with strain when the average stress on the cross section is lower than 0.4*f_ck_*; after that, the curve becomes nonlinear. Hence, the initial point of concrete plasticity with uniaxial compression was located at 0.4 times the standard value of concrete stress strength, i.e., at *σ = *0.4*f_ck_*. The concrete parameters of the plastic damage constitutive model (CDP) for C40 concrete were as shown in Table 3.

### 2.3. Finite Element Model

The axial impact was simulated by a rigid body with given quality and speed. At first, a concrete-filled steel tubular member bearing static axial load was calculated in ABAQUS/Standard. Then, the static calculation results in the restart document were imported into ABAQUS/Explicit to perform the calculation of a concrete-filled steel tubular member impacted with static axial load. 

The core concrete and steel tube were simulated using reduced integrated solid element C3D8R, and the impact block was simulated using rigid solid element R3D4. Hourglass Control was selected to analyze the grid accuracy. Sweep meshing was used to mesh for the finite element model. The meshing size of the concrete-filled steel tubular was about 0.01 and that of the impact block was about 0.04 to save calculation time.

It was assumed that there is no slip between the core concrete and steel tube; therefore, concrete elements and steel elements with the same geometric location shared the same node for the finite element calculation of the concrete-filled steel tubular member. 

For the contact surface between the impact block and the end of the member which bears the impact, a friction contact penalty function was used in the tangential direction, and 0.1 was adopted as the friction coefficient; hard contact was used in the normal direction. 

One end of the member was free and the other was fixed. 

Using the above finite element model, the whole impact process of the concrete-filled steel tubular member with axial compression bearing an axial impact load was simulated, and energy analysis and result verification was conducted. It showed that the meshing and contact parameters in this finite element model were suitable for dynamic analysis; the hourglass effect was very small and can be ignored. 

## 3. Results and Discussion

“Impact load” refers to the interaction force produced on the contact surfaces where the impact block and concrete-filled steel tubular member collide with each other. In this paper, three representative concrete ages, i.e., 28 days, 6 months, and 30 years, were chosen to evaluate various factors’ effects on the impact load. The time history curve of the impact load started at 1 s and ended at 1.005 s along the horizontal axis because axial static calculation was conducted during the first 1 s.

### 3.1. Impact Speed Effect on Impact Load

Impact speed is one of the key factors affecting the dynamic responses of a concrete-filled steel tubular member. The speed *v* was set to 5 m/s, 10 m/s, and 15 m/s to study its effect on impact load. The results are shown in Figure 2. 

From the time history curve of the impact load shown in Figure 2a–c, we can see that the impact load curves under various concrete ages maintained the same trend with different impact speeds. The impact load increased quickly when the impact block hit the member at low speed, then dropped down to zero when the block separated from the member; next, the second collision happened and unloaded in the end. When the impact speed was higher, after the first collision, the impact load increased quickly to the peak and then decreased rapidly to a plateau value because plastic deformation occurred in the member. Under instantaneous peak loads, the members easily enter plastic deformation. According to the impact load vs. time curves in this study, the peak impact load was significantly higher than the plateau value no matter the concrete age, which has a greater impact on the state of members. Therefore, in this study we focused on the peak impact load.

From Figure 2d, we can see that the peak impact load rose with increasing impact speed at the same concrete age. As the concrete age increased, the peak impact load dropped down at lower impact speed, for example, at 5 m/s and 10 m/s, while the peak impact loads were almost the same at higher impact speed; for example, when the impact speed was 15 m/s, the peak impact loads at different concrete ages, i.e., 28 days, 6 months, and 30 years, were 4036 kN, 4006 kN, and 3875 kN, respectively. At higher impact speed, the effect of creep on the peak impact load is relatively small.

### 3.2. Mass Ratio Effect on Impact Load

The mass ratio *x* was set to 0.5, 1, and 2 to study its effect on the impact load, and the impact velocity was kept constant at 5 m/s. The results are shown in Figure 3.

From the time history curves of impact load shown in Figure 3a–c, we can see that the impact load curves under various concrete ages maintained the same trend with different mass ratios. The impact load increased quickly with lower mass ratio when the impact block hit the member, then it dropped down to zero when the block separated from the member. After that, a secondary collision occurred between the member and the impact block. With higher mass ratio, the impact load increased rapidly when the first collision happened, then it fluctuated and gradually decreased at the end without a second collision. The contact time was longer when the mass ratio was bigger. The kinetic energy of the impact block was larger when the mass ratio was bigger, and the member showed plastic energy dissipation capacity during the impact process. 

From Figure 3d, we can see that the peak impact load slightly rose with increasing mass ratio at the same concrete age. For the same mass ratio, the peak impact load after core concrete creep occurring was far less than that without creep. The peak impact load at 28 days was 1.8 times that at 30 years, while the peak impact loads at 6 months and 30 years were almost the same, which is in accordance with the concrete creep rule. 

### 3.3. Axial Pressure Ratio Effect on Impact Load

The axial pressure ratio *n* was set to 0.3, 0.45, and 0.6 to analyze the effect on impact load. The results are shown in Figure 4.

From the time history curves of impact load shown in Figure 4a–c, we can see that the impact load curves maintained the same trend with different axial pressure ratios after creep occurred. When the axial pressure ratio was 0.3 or 0.45, the impact load increased quickly after the first collision of the impact block and the member, then dropped to zero because the block separated from the member. After that, they collided again. When the axial pressure ratio was 0.6, the impact load also increased rapidly and then dropped without the second collision. The results also show that the greater the axial pressure ratio, the longer the contact time between the impact block and the member. 

From Figure 4d, we can see that the peak impact load decreased with increasing axial pressure ratio when the core concrete age was 28 days. The reason for this is that the maximum core concrete stress reached 44.58 MPa and 50.15 MPa, respectively, for the members with axial pressure ratios of 0.45 and 0.6 when the collision happened. These stress values are greater than the ultimate compressive strength of the core concrete, so the core concrete was destroyed, and the impact resistance property of the whole member was weakened. However, for the members with core concrete ages 6 months and 30 years, the peak impact load increased with increasing axial pressure ratio. The reason for this is that the internal force of the section was redistributed, the stress of the steel tube section increased, and the core concrete section stress decreased. The core concrete section stress level was relatively lower when the member bore impact, and it did not exceed the ultimate compressive strength; increasing the axial pressure ratio had a potentiation effect on the impact resistance property of the member within certain limits.

### 3.4. Steel Ratio Effect on Impact Load

The steel ratio *α_s_* was set to 0.063, 0.108, and 0.156 to study its effect on impact load. The steel ratio was changed by changing the value of the wall thickness of the steel tube. The results are shown in Figure 5.

From the time history curves of impact load shown in Figure 5a–c, we can see that the impact load curves under various concrete ages maintained the same trend with different steel ratios. When the steel ratio was smaller, after the first collision of the impact block and the member, they separated from each other, then collided again, and a second peak impact load came into being. When the steel ratio was 0.156, after the first collision, the impact load increased quickly to the peak, then decreased rapidly; the second small peak came into being and then unloaded without a second collision. The larger the steel ratio, the shorter the contact time between the impact block and the member. The reason for this is that the bigger the steel ratio, the larger the member stiffness.

From Figure 5d, we can see that the peak impact load rose with increasing steel ratio at the same concrete age. The reason for this is that the bigger the steel ratio, the larger the member stiffness and the better the impact resistance of the member. At the same steel ratio, the peak impact load decreased with increasing concrete age, and it decreased quickly when the concrete age changed from 28 days to 6 months. This is because concrete creep develops faster during the early stage. 

### 3.5. Slenderness Ratio Effect on Impact Load

The slenderness ratio λ was set to 20, 40, and 60 to study its effect on impact load. The slenderness ratio was changed by changing the value of the member length of the steel tube. The results are shown in Figure 6.

From the time history curves of impact load shown in Figure 6a–c, we can see that the impact load curves under various concrete ages maintained the same trend with different slenderness ratios. After the first collision of the impact block and the member, they separated from each other and then collided again; the second peak impact load came into being and unloaded finally. At the same concrete age, after the first peak impact load, the bigger the slenderness ratio, the more slowly the impact load decreased to zero and the longer the contact time between the impact block and the member. The reason for this is that the bigger the slenderness ratio, the greater the flexibility of the member.

As shown in Figure 6d, the peak impact load rose with increasing slenderness ratio at the same concrete age. At the same slenderness ratio, the peak impact load decreased with increasing concrete age and it decreased quickly at early ages; this is also because concrete creep develops faster during the early stage.

### 3.6. Concrete Strength Effect on Impact Load

The concrete strength was set to C30, C40, and C50 to study its effect on impact load. The results are shown in Figure 7.

From the time history curves of impact load shown in Figure 7a–c, we can see that the impact load curves under various concrete ages maintained the same trend with different concrete strengths. After the first collision, they separated from each other and then collided again; the second peak impact load came into being and unloaded finally. At the same concrete age, the contact time between the impact block and the member changed little with changing concrete strength.

As shown in Figure 7d, the peak impact load rose with increasing concrete strength at the same concrete age. The reason for this is that the higher the concrete strength, the larger the concrete elastic modulus and the higher the member stiffness; hence, the better the impact resistance performance of the member. With the same concrete strength, peak impact load decreased with increasing concrete age, and it also decreased quickly at early ages.

### 3.7. Eccentricity of Axial Impact Effect on Impact Load

In order to research the impact position effect on the impact load of a concrete-filled steel tubular member under axial compression, three different impact positions were selected as shown in Figure 8. The distances between the centroids of impact blocks and the section centers of the members, called the eccentricity of axial impact in this paper, were 0 m, 0.10 m, and 0.141 m, respectively. The calculation results are presented in Figure 9.

From the time history curves of impact load shown in Figure 9a–c, we can see that the impact load curves under various concrete ages maintained the same trend with different impact positions. For Impact Position 3, after the collision of the impact block and member, the impact load increased rapidly and then decreased to a plateau, and the impact load was smaller. For Impact Positions 2 and 1, after the first collision, they separated from each other and then collided again; the second peak impact load came into being and unloaded finally.

As shown in Figure 9d, the peak impact load decreased with increasing eccentricity of the axial impact at the same concrete age. For the same impact position, the peak impact load decreased with increasing concrete age, and it decreased quickly at early ages. This is in accordance with the concrete creep rule.

### 3.8. Boundary Condition Effect on Impact Load

In order to study the boundary condition effect on the impact dynamic responses of a concrete-filled steel tubular member, four familiar boundary conditions were chosen for the calculation, as shown in Figure 10. The calculation results are presented in Figure 11.

From the time history curves of impact load shown in Figure 11a–c, we can see that the impact load curves under various concrete ages maintained the same trend with different boundary conditions. After the collision, the impact loads of the members with the first three boundary conditions were consistent. That of the member with the last boundary condition was smaller and the contact time was larger.

As shown in Figure 11d, the peak impact loads of the members with four different boundary conditions were almost equal at the same concrete age. The weaker the condition, the smaller the peak impact load. The peak impact load decreased quickly with increasing concrete age at the early stage, while it decreased slowly at greater concrete age.

## 4. Suggestions

### 4.1. Creep Reduction Coefficient on Peak Impact Load

In order to research the core concrete creep effect on the axial peak impact load of a concrete-filled steel tubular member, the above peak impact loads of the member under the variation of seven different parameters were analyzed, as shown in Table 4. The loading ratio given in the table was defined as the ratio of the peak impact load at different concrete ages to the initial value (in this paper, we mean the peak impact load at a concrete age of 28 days).

As shown in Table 4, for the different values of the eight parameters, the loading ratio was in the range of 0.51~0.86 when the concrete age was 6 months, its average value was 0.60, and the confidence interval for a 95% confidence coefficient was (0.549, 0.651). The loading ratio was in the range of 0.41~0.78 when the concrete age was 30 years, its average value was 0.55, and the confidence interval for a 95% confidence coefficient was (0.491, 0.600). 

### 4.2. Peak Impact Load Calculation Formula Considering Creep

The reduction coefficient for the peak impact load of a concrete-filled steel tubular member under axial compression considering creep was defined as the loading ratio in Section 4.1 and is expressed by *β*. From the above discussion for the eight selected parameters, we can see that the two peak impact load curves with the seven parameters at 6 months and 30 years were basically parallel. Linear interpolation was used to calculate the reduction coefficient for the peak impact load when the concrete age is between 6 months and 30 years. According to the simplified calculation formula for peak impact load in reference [9], the calculation formula for the peak impact load of a concrete-filled steel tubular member under axial compression considering creep can be expressed as
(1)$$P_{0}=\alpha \beta \frac{2mv}{T}$$
where *β* is the creep reduction coefficient for peak impact load (when the concrete age is 6 months, *β* = 0.60; when the concrete age is 30 years, *β* = 0.55; when the concrete age is between 6 months and 30 years, linear interpolation is used to calculate *β*); *α* is a correction factor for the impact load and is in the range of 1.93–2.3; *m* is the mass of the impact block; *v* is the impact speed; and *T* is the impact time. When the concrete age is 28 days, 6 months, or 30 years, the respective impact time calculation formulae based on the regression analysis can be obtained as follows:(2)$$T_{28\;days}\left(v,n,\alpha_{s},\lambda \right)=2.4\times 10^{-4}v-9.20\times 10^{-5}n+1.35\times 10^{-3}\alpha_{s}+1.05\times 10^{-5}\lambda -0.001\;T_{6\;months}\left(v,n,\alpha_{s},\lambda \right)=2.57\times 10^{-4}v+6.40\times 10^{-3}n+1.33\times 10^{-3}\alpha_{s}+1.30\times 10^{-5}\lambda -0.003\;T_{30\;years}\left(v,n,\alpha_{s},\lambda \right)=2.84\times 10^{-4}v+4.89\times 10^{-3}n+6.91\times 10^{-4}\alpha_{s}+2.25\times 10^{-5}\lambda -0.003$$
where *n* is the axial pressure ratio, *α_s_* is the steel ratio, and *λ* is the slenderness ratio.

Figure 12 presents a comparison of the results of Equation (1) and the simulation results for peak impact load. From Figure 12 we can see that the formula calculation results are in good agreement with the simulation results. When the concrete age is 28 days, 6 months, or 30 years, the peak values of impact load using Equation (1) are 2438.63 kN, 1199.81 kN, and 1108.77 kN, respectively, while those using FEM are 2484.31 kN, 1252.40 kN, and 1136.58 kN; the relative errors are thus 2%, 4%, and 3%, respectively.

## 5. Conclusions

In this paper, the whole process of a concrete-filled steel tube subjected to axial impact was calculated using the finite element method. The influences of various possible factors on the impact loads were analyzed in as much detail as possible. Concrete creep was also considered in the study. Through this research, the following conclusions can be drawn:(1)The equivalent elasticity modulus was introduced to simulate the dynamic responses of a concrete-filled steel tubular member under the combined action of axial compression, creep, and axial impact. The calculation results show that the finite element model established in this paper is reasonable, accurate, and suitable for the simulation.(2)The peak impact load and contact time increased with increasing impact speed; when the impact speed was higher, the effect of concrete creep on the impact load was weakened.(3)The impact load did not change much with increasing mass ratio, while contact time increased with increasing mass ratio. For the same mass ratio, the peak impact load after core concrete creep occurring was far less than that without creep.(4)After concrete creep occurring, the peak impact load rose with the axial pressure ratio, steel ratio, slenderness ratio, and concrete strength. The axial pressure ratio, within certain limits, had a potentiation effect on the impact resistance of the member.(5)With increasing eccentricity of the axial impact, the peak impact load decreased linearly. Enhancement of the constraints at the two ends is conducive to improving the impact resistance of the member.(6)The calculation formula constructed in this paper for the peak impact load of a concrete-filled steel tubular member under axial compression considering creep is easy to use and can help in the impact resistance design of concrete-filled steel tubular members.

## Figures and Tables

**Figure 1 materials-12-03134-f001:**
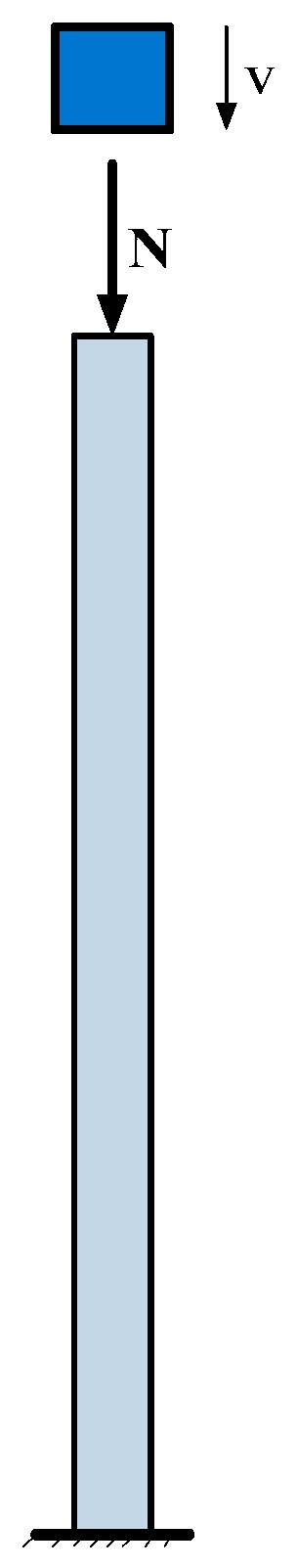
Geometric model.

**Figure 2 materials-12-03134-f002:**
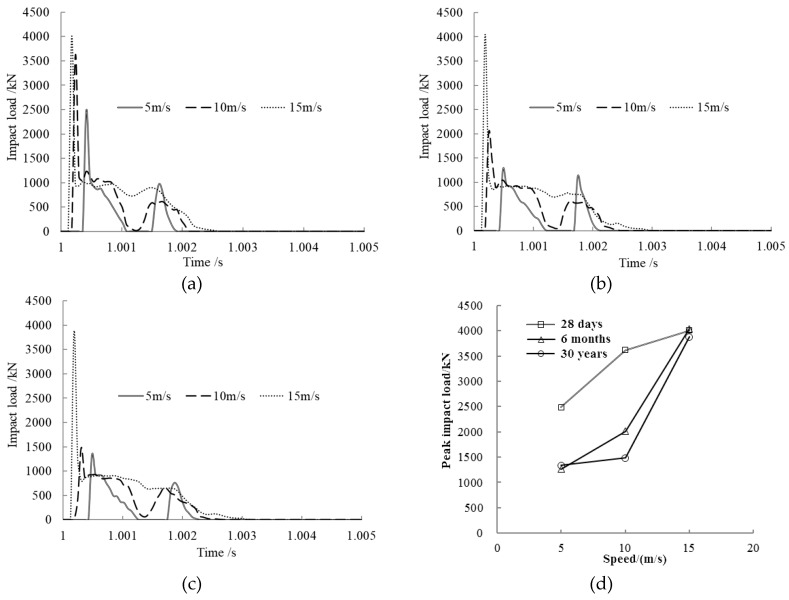
Impact speed effect on impact load. (**a**) Concrete age, 28 days; (**b**) Concrete age, 6 months; (**c**) Concrete age, 30 years; (**d**) Comparison of peak impact loads.

**Figure 3 materials-12-03134-f003:**
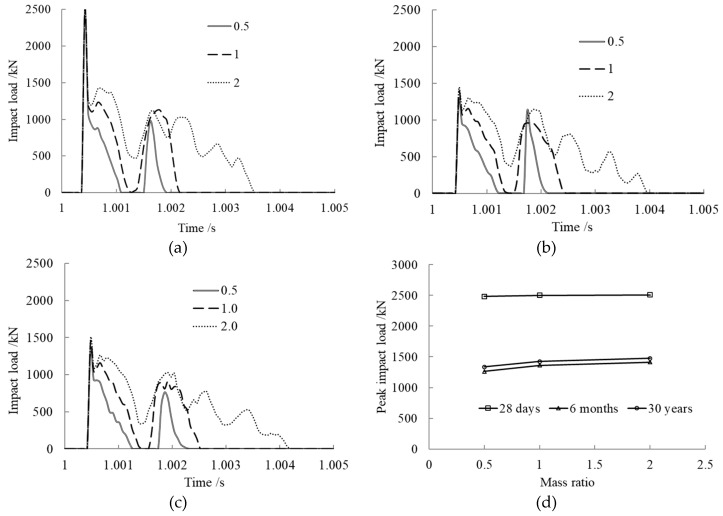
Mass ratio effect on impact load. (**a**) Concrete age, 28 days; (**b**) Concrete age, 6 months; (**c**) Concrete age, 30 years; (**d**) Comparison of peak impact loads.

**Figure 4 materials-12-03134-f004:**
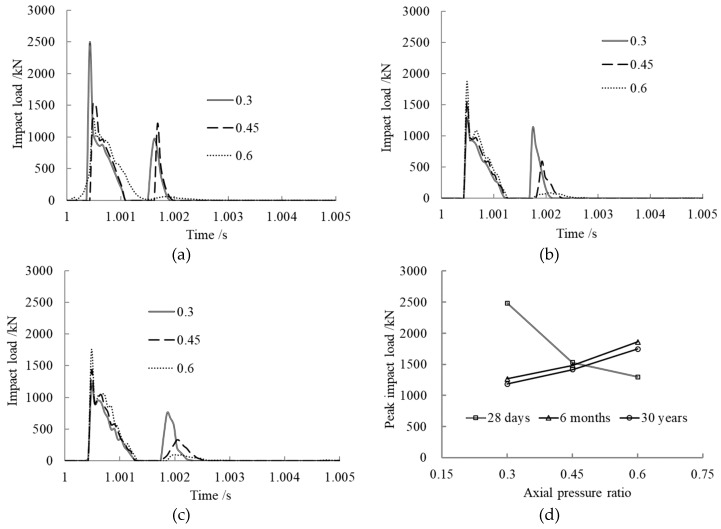
Axial pressure ratio effect on impact load. (**a**) Concrete age, 28 days; (**b**) Concrete age, 6 months; (**c**) Concrete age, 30 years; (**d**) Comparison of peak impact loads.

**Figure 5 materials-12-03134-f005:**
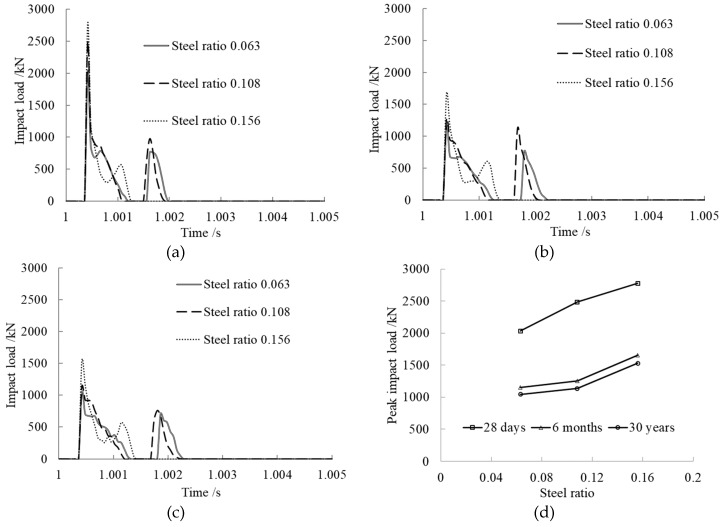
Steel ratio effect on impact load. (**a**) Concrete age, 28 days; (**b**) Concrete age, 6 months; (**c**) Concrete age, 30 years; (**d**) Comparison of peak impact loads.

**Figure 6 materials-12-03134-f006:**
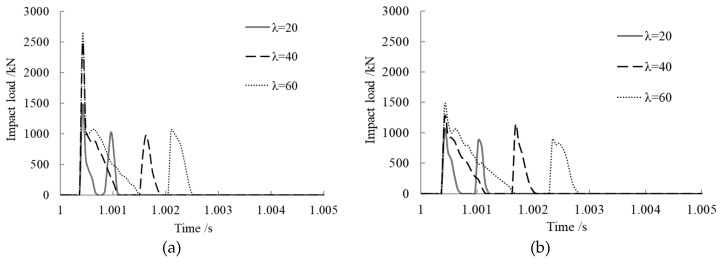
Slenderness ratio effect on impact load. (**a**) Concrete age, 28 days; (**b**) Concrete age, 6 months; (**c**) Concrete age, 30 years; (**d**) Comparison of peak impact loads.

**Figure 7 materials-12-03134-f007:**
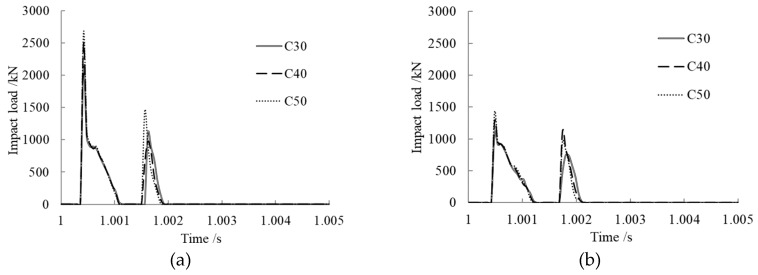
Concrete strength effect on impact load. (**a**) Concrete age, 28 days; (**b**) Concrete age, 6 months; (**c**) Concrete age, 30 years; (**d**) Comparison of peak impact loads.

**Figure 8 materials-12-03134-f008:**
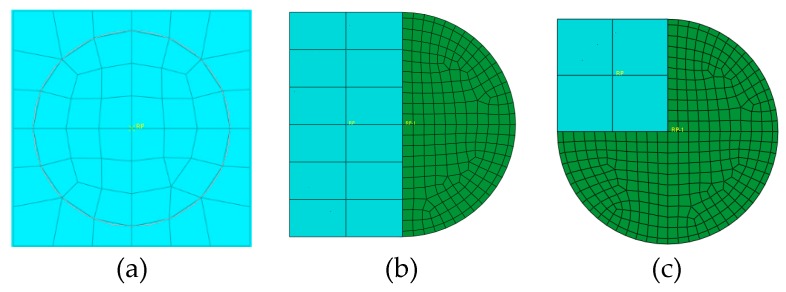
Three different impact positions. (**a**) Impact Position 1; (**b**) Impact Position 2; (**c**) Impact Position 3.

**Figure 9 materials-12-03134-f009:**
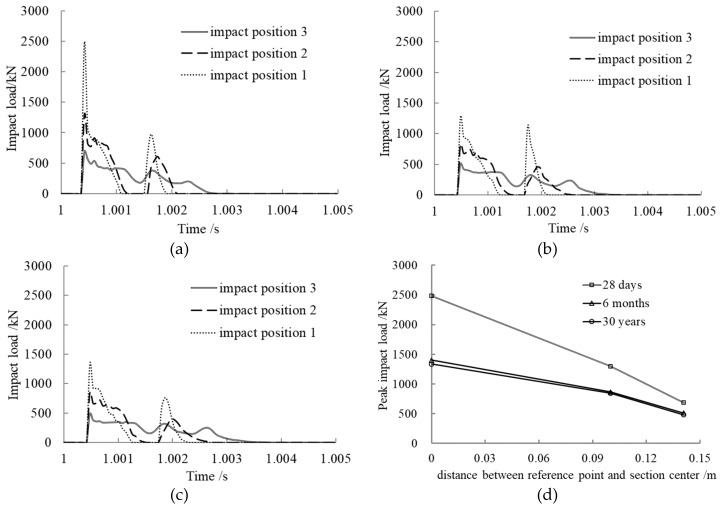
Impact position effect on impact load. (**a**) Concrete age, 28 days; (**b**) Concrete age, 6 months; (**c**) Concrete age, 30 years; (**d**) Comparison of peak impact loads.

**Figure 10 materials-12-03134-f010:**
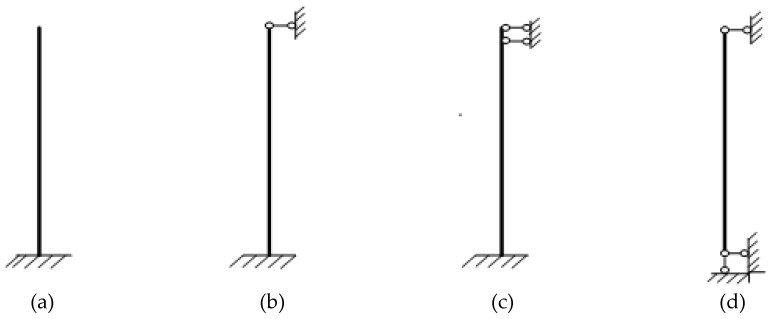
Four different boundary conditions. (**a**) One end fixed and the other free; (**b**) One end fixed and the other simply supported; (**c**) One end fixed and the other sliding support; (**d**) Two ends simply supported.

**Figure 11 materials-12-03134-f011:**
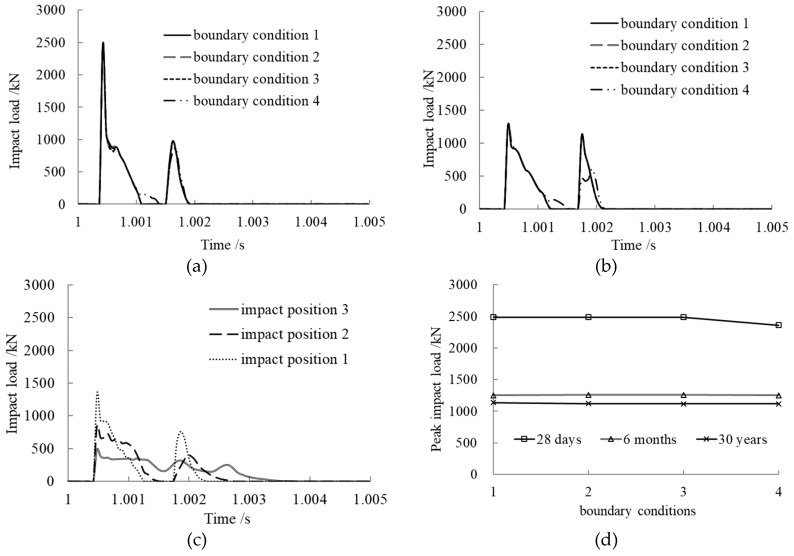
Boundary condition effect on impact load. (**a**) Concrete age, 28 days; (**b**) Concrete age, 6 months; (**c**) Concrete age, 30 years; (**d**) Comparison of peak impact loads.

**Figure 12 materials-12-03134-f012:**
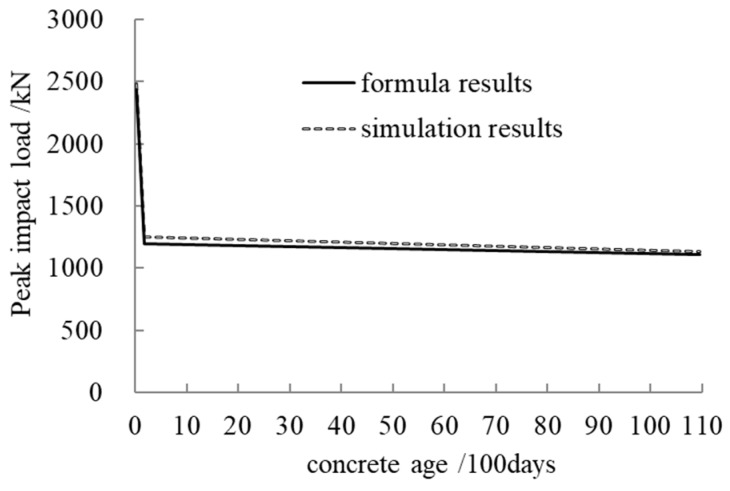
A comparison of peak impact load.

**Table 1 materials-12-03134-t001:** Detailed parameters.

*D*/mm	*t*/mm	*L*/mm	*α_s_*	*λ*	*N*/kN	*n*	*x*	*m*/kg	*v*/(m/s)
200	5	2000	0.108	40	423	0.3	0.5	90	5

**Table 2 materials-12-03134-t002:** Equivalent elasticity moduli of the core concrete.

Age	28 Days	6 Months	30 Years
Equivalent elasticity modulus/MPa	32,500.00	20,420.17	17,403.85

**Table 3 materials-12-03134-t003:** Concrete parameters in the plastic damage constitutive model.

Compressive Strength/MPa	Inelastic Strain (Compressive)	$\mathrm{Damage}\;\mathrm{Factor},\;d_{c}$	Tensile Strength/MPa	Inelastic Strain (Tensile)	$\mathrm{Damage}\;\mathrm{Factor},\;d_{t}$
10.72	0.000000	0.000	2.39	0.00000000	0.000
13.93	2.70 × 10^−5^	0.015	2.32	2.47 × 10^−5^	0.199
17.42	8.10 × 10^−5^	0.035	2.03	5.65 × 10^−5^	0.393
20.34	1.56 × 10^−4^	0.056	1.63	1.03 × 10^−4^	0.596
22.70	2.54 × 10^−4^	0.080	1.34	1.46 × 10^−4^	0.716
24.53	3.72 × 10^−4^	0.105	1.08	1.97 × 10^−4^	0.809
26.85	8.48 × 10^−4^	0.196	0.88	2.57 × 10^−4^	0.872
25.85	1.26 × 10^−3^	0.275	0.68	3.48 × 10^−4^	0.922
22.49	1.96 × 10^−3^	0.403	0.58	4.25 × 10^−4^	0.945
20.11	2.43 × 10^−3^	0.483	0.45	5.76 × 10^−4^	0.968
18.99	2.66 × 10^−3^	0.520	0.38	7.14 × 10^−4^	0.978
15.24	3.56 × 10^−3^	0.644	0.28	1.03 × 10^−3^	0.988
13.77	3.99 × 10^−3^	0.692	-	-	-
10.12	5.45 × 10^−3^	0.807	-	-	-
6.82	7.83 × 10^−3^	0.899	-	-	-
2.61	1.86 × 10^−2^	0.982	-	-	-

**Table 4 materials-12-03134-t004:** Creep effect coefficient on peak impact load.

Parameters	28 Days	6 Months		30 Years	
	Loading Ratio	Loading Ratio	Average Value	Loading Ratio	Average Value
Reference set	1.00	0.50	0.60	0.46	0.55
Impact speed *v* = 10m/s		0.56		0.41	
Impact speed *v* = 15m/s		0.99 ^1^		0.96 ^1^	
Mass ratio *x* = 1		0.57		0.53	
Mass ratio *x* = 2		0.59		0.56	
Axial pressure ratio *n* = 0.2		0.68		0.65	
Axial pressure ratio *n* = 0.6		0.86		0.78	
Steel ratio *α_s_* = 0.063		0.57		0.51	
Steel ratio *α_s_* = 0.156		0.60		0.55	
Slenderness ratio *λ* = 20		0.72		0.66	
Slenderness ratio *λ* = 60		0.55		0.47	
Concrete strength *C30*		0.52		0.47	
Concrete strength *C50*		0.53		0.50	
Impact position 2		0.74		0.70	
Impact position 3		0.66		0.65	
Boundary condition 2		0.51		0.45	
Boundary condition 3		0.51		0.45	
Boundary condition 4		0.53		0.47	

^1^ The maximum value was rejected when we calculated the average value of the loading ratio due to little effect of higher impact speed on the peak impact load, as shown in Section 3.1.

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
