# Peer review of "Axial Impact Load of a Concrete-Filled Steel Tubular Member with Axial Compression Considering the Creep Effect"

_materials, 2019, doi:10.3390/ma12193134_

Round 1

Reviewer 1 Report

line 86 What does Es mean ? - the value 2.1 GPa looks strange.

line 91 the factor 0.4 needs deeper explanation 

The used concrete model with the creep effect should be wider described.

line 114-115 I would expect some more details of the mesh,  especially the ref. 17 is in Chinese.

Impact speed effect: the difference between the values for 6 months and for 30 years with the speed 10 m/s is not explained. 

The results are interesting but the paper needs some improvements e.g. the used material models and FEM mesh should be described. The finale conclusions could be focused on the used calculation method and its advantages and disadvantages, not only on the results.

Reviewer 2 Report

The manuscript needs major revision as the contribution of the author (over and above the literature review) is not very visible. Language has to be significantly improved. The sentences are lengthy and at some instances run to four full lines. Technical terms are also confusingly used. At most places the usage is a complete blunder. These two are the major limitations which do not encourage reading at all. I would recommend outright rejection of this paper. Secondly this paper seems to be a just a report of technical investigation. There are no reasons given as to why these results have been obtained, nor is there any comparison to previous work in the same or similar domain. The references are also limited to one geographic region.

The presentation of the manuscript abounds in disjointed English, lacks coherence and clarity. Choice of words also needs substantial improvement. Sentence structuring is below average.

The list could go on, but the bottom line is that the authors need to rewrite the paper, or even reconsider the research content, before it could be considered for publication in this journal. The manuscript  hovers around 'rejection'. 

Round 2

Reviewer 1 Report

Majority of my doubts are clarified, but the way of  factor 0,4 calculation should be written down in the work, I am not able to find and read from [17]. The literature should be wider. The FEM analysis description is too narrow and needs more details - wider qualitative and quantitative description.  The English should be improved. 

Reviewer 2 Report

The authors have improved the paper in better manner. Still scientific value and the contribution points need to be projected well in the manuscript. The significance of content should be focused well so that the overall content will ne improved.

Many mistakes can be encountered even in abstract,

"The fruits provide a scientific basis for the impact resistance design of concrete-filled steel tubular member."

In academic writings, the word 'fruit' cannot be used. Change the word phrase and looking carefully into the entire manuscript content.

After incorporating all the changes, the manuscript can be accepted. I feel it will take really good amount of time to carefully improve the manuscript. If possible take any English service for correcting the paper.
